# Microbial Degradation of Chromium-Tanned Leather During Thermophilic Composting: A Multi-Scale Analysis of Microbial Communities and Structural Disruption

**DOI:** 10.3390/biology14121799

**Published:** 2025-12-18

**Authors:** Manuela Bonilla-Espadas, Irene Lifante-Martinez, Mónica Camacho, Elena Orgilés-Calpena, Francisca Arán-Aís, Marcelo Bertazzo, María-José Bonete

**Affiliations:** 1Footwear Technological Centre, INESCOP, 03600 Elda, Spain; mbonilla@inescop.es (M.B.-E.); ilifante@inescop.es (I.L.-M.); eorgiles@inescop.es (E.O.-C.); aran@inescop.es (F.A.-A.); 2Grupo Biotecnología de Extremófilos, Departamento de Bioquímica y Biología Molecular y Edafología y Química Agrícola, Universidad de Alicante, 03690 San Vicente del Raspeig, Spain; camacho@ua.es

**Keywords:** chromium-tanned leather, composting, biodegradation, thermophilic microorganisms, microbial communities, biofilms

## Abstract

Leather waste, particularly that derived from chromium-tanned leather that generated from chromium-tanned leather, is difficult to manage because it contains both persistent collagen and chromium compounds that can be harmful if not handled properly. In this work, we used thermophilic composting as a controlled laboratory tool to establish a reproducible system in which small pieces of chromium-tanned leather were exposed to the naturally occurring microbiota and the high temperatures typical of this process. Our aim was not to certify that this process already offers a safe final solution for leather waste, but to create a reproducible thermophilic composting environment suitable for characterising the microbial communities that develop in compost in contact with chromium-tanned leather and for isolating some of their members for further research. We combined small (2 L) and larger (40 L) composting systems with DNA sequencing, microscopy and classical microbiology to examine how thermophilic composting conditions affected the leather material and the associated microbiota. The composting conditions led to a marked alteration of the leather structure. They identified bacterial and fungal groups adapted to warm, chromium-rich environments, from which we obtained several cultivable strains. These results provide a first microbiological basis for developing future biotechnological tools to manage chromium-tanned leather waste more safely. However, additional studies are still needed before this approach can be recommended as a routine treatment option in real waste management scenarios.

## 1. Introduction

The management of solid leather waste, particularly from chromium-tanned (wet-blue) leather, remains a persistent challenge for both the leather industry and environmental authorities [1]. Leather production generates large volumes of organic waste [2], with hides and skins subjected to tanning processes that confer durability and resistance to microbial degradation, but that also hinder their reintegration into biological cycles [3]. This waste primarily originates from the tanning stage, which consumes significant amounts of water and energy and involves the use of various chemicals [4]. Chromium(III) salts are widely used in tanning due to their affinity for cross-linking with collagen [5]. One of the main environmental concerns is the use of chromium salts in tanning, as trivalent chromium Cr(III) can oxidise to its hexavalent form Cr(VI), which is highly toxic [6]. This transformation poses a serious environmental hazard, particularly when waste is mismanaged or improperly disposed of, leading to potential contamination of soil and water bodies [7].

Although various techniques for recycling and resource recovery from leather waste are being investigated, several limitations still hinder their scalability and broader implementation. For instance, leather waste has been successfully blended with natural rubber to produce composite materials with enhanced mechanical properties, including increased tensile strength and hardness. These composites show potential for use in the manufacture of shoes, bags, and upholstery products [8,9]. However, these applications often reduce the quality and recyclability of the resulting products, limiting their long-term circularity. In terms of chemical recovery, extraction methods for chromium include acid, alkaline, or enzymatic hydrolysis, sometimes assisted by organic chelators or ultrasound, aiming to reduce the environmental impact of chromium-laden waste [10]. Additionally, the recovery of collagen and related protein fractions for reuse in biomaterials or industrial formulations is being explored [11]. These approaches remain at an early stage of development and have not yet been implemented on an industrial scale [12,13]; consequently, conventional disposal methods, such as landfilling or incineration, continue to be used, despite raising serious environmental and regulatory concerns [14].

Composting is a well-established biological treatment method for organic waste, relying on aerobic microbial metabolism and thermophilic conditions to convert substrates into stable humic products [15]. However, its application to chromium-tanned leather waste remains largely unexplored, mainly due to the presumed toxicity of chromium salts and the structural complexity of the tanned collagen matrix [16,17]. The tanning process involves the cross-linking of collagen fibres with trivalent chromium, which enhances durability but significantly reduces biodegradability by inhibiting enzymatic activity and microbial colonisation [18,19]. While the partial biodegradability of untanned or vegetable-tanned leather has been demonstrated under composting or anaerobic digestion [20,21,22], the microbial processes specifically involved in the degradation of chromium-tanned leather remain poorly understood [23]. Some chemical and physical treatments, including acid, alkaline, and enzymatic hydrolysis, sometimes assisted by ultrasound, have been explored for chromium recovery; however, their efficiency and scalability remain limited [10]. In parallel, the chemical reactivity of chromium-tanned leather has been studied through its dye-adsorption capacity, providing insights into mass transfer mechanisms during degradation [24]. Additionally, archaeological evidence of microbial decay in vegetable-tanned leather provides valid reference points, although the behaviour of chromium-tanned leather under similar conditions remains insufficiently characterised [25].

In line with the growing interest in circular economy strategies for treatment of tanned leather waste [26,27,28], recent efforts have focused on understanding the microbial and biochemical mechanisms that enable its biodegradation under controlled conditions. Microbial consortia in composting play a pivotal role in degrading complex organic materials into simpler compounds, adapting to environmental stressors, and harbouring enzymatic potential for degradation processes [29]. These consortia comprise diverse microorganisms, including bacteria, fungi, and archaea, each contributing to the breakdown of organic matter during the composting phases [30]. Composting involves a mesophilic phase followed by a thermophilic phase, each dominated by distinct microbial communities [31]. Mesophiles initiate the degradation of simple compounds, while thermophiles take over at higher temperatures to break down more complex materials [32]. Microbial consortia exhibit resilience and adaptability to environmental stressors, such as metal pollution, by developing tolerance mechanisms [33]. These mechanisms include the formation of efflux systems and enzymatic detoxification, which enable the consortia to adapt and survive in contaminated environments, demonstrating the consortium’s potential for environmental remediation [34]. Moreover, the role of biofilm formation and extracellular polymeric substances in enhancing microbial adhesion, enzymatic synergy, and metal tolerance has become increasingly recognised [35]. Furthermore, simplifying microbial consortia through strategies such as enrichment and dilution can maintain or even enhance their degradation capabilities, underscoring the potential to develop engineered consortia for specific applications in waste management and bioremediation [36].

This study is based on the hypothesis that thermophilic composting can enable the microbial degradation of chromium-tanned leather and promote the selection of microbial taxa capable of degrading collagen and tolerating chromium. This study does not seek to evaluate the compostability of chromium-tanned leather according to standardised criteria or to validate thermophilic composting as a ready-to-implement disposal option. Instead, our main objective was to establish a reproducible thermophilic composting system in which chromium-tanned leather is exposed to realistic physicochemical and microbiological conditions, allowing (i) multi-scale characterisation of the associated bacterial and fungal communities and (ii) isolation of representative cultivable strains. To this end, we combined small-scale (2 L) and pilot-scale (40 L) composting assays with high-throughput amplicon sequencing, culture-dependent isolation and microscopic and gravimetric analyses of the impact of composting conditions on leather structure. The knowledge generated provides a microbiological basis for future strategies aimed at enzymatic degradation, safe valorisation or improved management of chromium-tanned leather waste within circular bioeconomy frameworks.

## 2. Materials and Methods

### 2.1. Thermophilic Composting Setups and Operation

Composting was performed using wet blue leather, a semi-processed material obtained through chromium tanning, which imparts a characteristic blue colour and enhances mechanical and thermal stability. This type of leather retains high moisture content and requires further processing for end-use applications [22,37,38]. The composting system employed in this study was designed following the principles outlined in ISO 16929:2021 [39], which defines thermophilic composting conditions for organic materials. However, the objective here was not to evaluate disintegration according to standardised criteria, but rather to establish a controlled composting environment suitable for microbiological identification and isolation. Two composting systems were employed: small-scale composting (using 2 L Dewar vessels) and large-scale composting (using a 40 L Dewar vessel) (KGW-Isotherm, Karlsruhe, Germany) (Figure 1 and Figure 2). For the small-scale setup, two parallel vessels were prepared (Figure 1), each containing a mixture of hay (40%), plant residues (30%) and manure (30%). The hay consisted of chopped cereal straw, the plant residues were mainly green pruning waste from horticultural and ornamental plants, and the manure fraction corresponded to a mature livestock manure compost routinely produced at a local agricultural facility. These ingredients were selected to provide a balanced carbon-to-nitrogen ratio, with the initial mixture formulated to fall within the C/N range of approximately 20–30, which is generally recommended for manure-based composting systems [40]. One vessel served as a control, while the second included three leather samples, 7 × 1.5 cm, shown in Figure 3B. The initial weights of the leather samples were determined using an ultra-precision balance (Appendix A). Replicates were obtained by placing three individual leather pieces within each composting vessel rather than using separate containers. This approach ensured consistent composting conditions for all replicates within each system while allowing for triplicate sampling for analysis. For the large-scale setup, the composting matrix contained the same type of cereal straw hay and livestock manure, mixed with mature compost from the small-scale system (10%), which acted as an inoculum rich in thermophilic microorganisms and helped stabilise moisture and nutrient availability (Figure 2). Three larger wet blue leather samples shown in Figure 3B were composted.

Initial weights were determined using an ultra-precision balance (Appendix A). The composting process lasted 44 days (1056 h) and was performed under thermophilic conditions (≥50 °C), with temperature continuously monitored throughout. Reinoculation and aeration were carried out only when the temperature dropped below 55 °C. This involved removing the composted material from the vessel, adding fresh plant residues, hay, and manure, and thoroughly mixing to restore microbial activity and oxygen availability. It is important to note that temperature changes in composting systems were not externally controlled but arose from microbial metabolic activity. As readily degradable organic substrates are consumed, microbial activity tends to decrease, leading to a natural drop in temperature. To restore thermophilic conditions and maintain active composting, fresh organic material was reinoculated. Although no dedicated control vessel was used in the large-scale (40 L) composting system, compost samples were collected at time zero, before the addition of leather, and used as a baseline reference for microbial composition. This approach allowed comparative analyses while preserving the internal thermodynamic dynamics of the composting process. Each composting system (2 L and 40 L) consisted of a single Dewar vessel, and sampling was performed in triplicate from each unit. This design was selected to ensure environmental consistency within each system and is consistent with proof-of-concept strategies used to generate preliminary data under controlled composting conditions as summarised in Figure 1 and Figure 2.

### 2.2. Metagenomic DNA Extraction and Sequencing

After 44 days of composting, a total of 10 samples (ine17 to ine26, see Table 1) were collected for DNA extraction. Strains were given internal collection codes of the form “INE-XXXX”, where “INE” refers to the INESCOP strain collection and the numerical part is a unique identifier without further experimental meaning. The numbering of samples starts at 17 because the present study is a continuation of previous experimental work in the same composting line, where samples 1–16 were used in a separate assay already published in [41]. This approach was adopted to maintain traceability and consistency in the documentation of biological materials. These included both bulk compost material and surface-scraped material from leather samples obtained from the 2 L and 40 L composting systems. For leather-associated samples, the leather surface was gently scraped with a sterile scalpel to recover solid material containing microbial biomass. All samples were homogenised and resuspended in the lysis buffer provided in the NZY Soil gDNA Isolation Kit (NZYtech, Lisboa, Portugal). DNA extraction was then performed according to the manufacturer’s protocol, which includes chemical lysis, mechanical disruption via bead beating, and purification using silica membrane-based spin columns [42]. Metagenomic sequencing was conducted on the Illumina MiSeq platform, focusing on the 16S rRNA gene for bacterial identification [43] and the Internal Transcribed Spacer (ITS) regions of the 23S rRNA gene for fungal identification [44].

### 2.3. Bioinformatics and Statistical Analysis

Amplicon libraries targeting the bacterial 16S rRNA and fungal ITS regions were prepared according to Illumina’s standard protocol and sequenced on an Illumina MiSeq platform (M-GL-00006 v1.0 ESP) (2 × 300 bp). The complete description of the amplification and library preparation protocol can be found in [45]. The raw Illumina sequences were imported into the Qiime2 (2024.5) bioinformatics tool [44] to perform initial quality control using DADA2 (1.26.0). Taxonomic assignment of each amplicon sequence variant (ASV), defined at a 99.9% sequence similarity, was performed using the classify-Sklearn module in combination with the SILVA v138 database [46] for bacteria and UNITE v8.2 for fungi. Statistical and microbial ecology analyses were performed using various R 4.3.1 software packages, including Phyloseq (1.44.0) [47] and Vegan (2.6-4) [48]. Alpha diversity was measured using the richness, Shannon diversity index, and Simpson diversity index. To determine significant differences in the relative abundances of taxa, the DESeq2 (1.36.0) test [49] was used. Beta diversity was assessed using Principal Coordinates Analysis (PCoA), based on a Bray–Curtis dissimilarity matrix [50]. PCoA is an ordination method used to evaluate the similarity of microbial communities. The PERMANOVA test was used to determine statistically significant differences between groups [51].

### 2.4. Cultivable Bacterial and Fungal Species Identification

In addition to metagenomic analysis, compost samples were plated on various growth media Tryptic Soy Broth media (TSB) (Condalab, Madrid, Spain) 3% and 1.5% agar, Reasoner’s 2A agar media (R2A) [peptone 0.1%; yeast extract 0.05%; dextrose 0.05%; starch 0.05%; K_2_HPO_4_ 0.03%; Mg_2_SO_4_·7H_2_O 0.005%; sodium pyruvate 0.03%], Nutrient Agar Isolate media (NAI; peptone 1%; meat extract 0.5%; NaCl 0.5%; 1.5% agar; pH 7.2) and Yeast Mould media (YM; yeast extract 0.3%; malt extract 0.3%; soybean peptone 0.5%; glucose 1%; agar 1%) to identify cultivable bacterial and fungal strains. Identifications were made by partial sequencing of the *16S rRNA* gene for bacteria and the *ITS* region for fungi, using universal primers ITS1 and ITS4. Sequences were analysed using the National Centre for Biotechnology Information NCBI Basic Local Alignment Search Tool (BLAST, 2.13.0 and 2.16.0) to determine the closest known strains [52]. Phylogenetic trees (dendrograms) based on *16S rRNA* and *ITS* sequences were constructed to support taxonomic assignment. These are available in the Appendix A. All sequences are being submitted to GenBank, and accession numbers will be included once available.

### 2.5. Scanning Electron Microscopy (SEM) and Gravimetric Analysis: Leather Analysis

At the end of the composting period (44 days), leather samples from the small-scale system (2 L Dewar vessels) were recovered and analysed gravimetrically. Recovered fragments were photographed, then sequentially washed with 10% sodium dodecyl sulphate (SDS), 70% ethanol, and distilled water to remove adherent microbial biomass. The samples were then air-dried under controlled laboratory conditions until they reached a constant weight. Final dry weights were measured using an ultra-precision balance. Gravimetric degradation was expressed as the percentage mass loss relative to each sample’s initial dry weight before composting (see Appendix A). In contrast, leather samples from the large-scale system (a 40 L Dewar vessel) were not suitable for gravimetric analysis due to advanced degradation; only fragmented pieces smaller than 1 cm could be recovered. These remains were instead analysed using Scanning Electron Microscopy (SEM), following the methodology described by Vyskočilová et al. [53], to assess surface morphology, structural degradation, and potential biofilm formation. Additionally, untreated leather samples were analysed using SEM before composting as a structural reference. These control samples were examined under the same SEM conditions to document the leather’s native ultrastructure and fibrillar integrity before microbial exposure.

## 3. Results

### 3.1. Temperature Monitoring During Thermophilic Composting

In both composting setups, self-heating rapidly led to thermophilic conditions. In the small-scale system (2 L vessels), temperatures reached ≥50 °C within the first 48 h and several peaks above 60 °C were recorded (Appendix A). Occasional additions of fresh compost were applied when temperatures dropped, ensuring that thermophilic conditions were re-established and maintained throughout the assay. In the larger system (40 L vessel), the greater thermal mass resulted in a more stable profile, with maximum temperatures above 70 °C and fewer reinoculations needed to sustain elevated temperatures (Appendix A). Overall, the temperature regimes achieved in both systems were consistent with those typically reported for thermophilic composting and indicate that the process proceeded under active thermophilic conditions suitable for microbiological characterisation.

### 3.2. Microbial Characterisation: Bacterial Community Composition

#### 3.2.1. Rarefaction, Alpha and Beta Diversity

Rarefaction curves reached saturation across all samples, indicating that sequencing depth was sufficient to capture the bacterial diversity present (Appendix A). Alpha diversity analysis at the amplicon sequence variant (ASV) level showed that compost control samples exhibited substantially higher diversity than leather compost samples. Specifically, observed ASV richness exceeded 680 in control samples (ine19 and ine20), while leather compost samples (ine17 and ine18) showed values below 400 (Appendix A). Similarly, the Shannon and Simpson indices were consistently higher in the control compost, with Shannon values above 5.1 and Simpson values near 0.99, compared to values below 4.5 and 0.97, respectively, in the leather compost. Beta diversity analysis, using Bray–Curtis dissimilarity and Principal Coordinates Analysis (PCoA), revealed clear compositional separation between sample types in both composting systems (Appendix A). The percentages of variance indicated in the PCoA plots correspond to the variance explained by Axis 1 alone, which captures the highest proportion of compositional differences between the samples. In the small-scale system, control samples (ine19, ine20) clustered tightly on the right side of Axis 1 (97.7% of explained variance), while leather compost samples (ine17, ine18) grouped on the left. A similar pattern was observed in the large-scale system: control samples (ine21–ine23) clustered on the left side of Axis 1 (65% variance), and leather compost samples (ine24–ine26) formed a distinct cluster on the right, with substantial intra-group similarity in both cases. Differential abundance analysis between the small-scale samples (ine17–ine20) revealed that compost control samples (ine19 and ine20) had significantly higher abundances for 124 bacterial genera. In contrast, only 49 genera were more abundant in the leather compost samples (ine17 and ine18). Overall, compost control samples contained 196 more genera than leather compost samples. Although differences in alpha diversity indices were evident between sample types, no statistical tests were applied due to the limited number of biological replicates per group. Therefore, diversity patterns were interpreted descriptively, which is appropriate for exploratory analyses of microbial communities.

#### 3.2.2. Taxonomic Distribution at Phylum and Genus Levels

In samples ine17 to ine20, the taxonomic classification at the phylum level revealed high similarity across all replicates. The most abundant phyla across both control and leather compost samples were *Firmicutes, Proteobacteria*, and *Bacteroidota. Actinobacteria* were also present at moderate levels across samples, particularly in leather compost (Figure 4; Appendix A).

At the genus level, more evident differences between control compost and leather compost samples were observed (Figure 5; Appendix A). *Puia* was more abundant in leather compost samples, whereas *Ruminofilibacter* dominated the control compost and was nearly absent in leather compost.

Genera such as *Sinibacillus*, *Proteiniborus*, *Desulfotomaculum*, *Chelativorans*, and *Bdellovibrio* were notably associated with leather compost samples, while *Ruminofilibacter* [54] was more frequent in the control group.

In samples ine21 to ine26, taxonomic profiles at the phylum level also showed substantial similarity between replicates. *Firmicutes* and *Actinobacteria* were generally more abundant in leather compost samples, while *Bacteroidota* appeared more frequently in control compost (Figure 6; Appendix A).

At the genus level, more distinct differences between control and leather compost samples were again evident (Figure 7; Appendix A). Genera such as *Novibacillus*, *Aeribacillus*, and *Thermobifida*, previously described in high-temperature composting contexts [55,56,57], were detected exclusively in leather compost samples. In contrast, genera such as *Ruminofilibacter*, also present in the small-scale control compost samples, were dominant in the control group and absent in the leather samples. *Galbibacter*, *Brumimicrobium*, and *Providencia* were statistically more abundant in the leather compost, whereas *Ruminiclostridium*, *Petrimonas*, and *Herbinix* were statistically more abundant in the control compost.

### 3.3. Microbiota Characterisation: Fungal Community Dynamics

#### 3.3.1. Rarefaction, Alpha and Beta Diversity

Rarefaction curves at the Operational Taxonomic Unit (OTU) level reached saturation across all samples, indicating that sequencing depth was sufficient to capture fungal diversity (Appendix A). Alpha diversity analysis based on ASV-level metrics revealed variable patterns across sample types. In the small-scale system using the 2 L Dewar vessel (samples ine17 to ine20), observed ASV richness was higher in the control compost samples (ine17 and ine18), while Shannon and Simpson indices were higher in the leather compost samples (ine19 and ine20) (Appendix A). In the large-scale 40 L Dewar vessel (samples ine21 to ine26), observed richness was similar across all samples. However, the Shannon and Simpson indices remained consistently higher in the leather compost group than in the control (Appendix A). Beta diversity analysis, using Bray–Curtis dissimilarity and PCoA, revealed a clear separation between leather compost and control compost samples in both composting systems, with strong similarity among replicates within each group (Appendix A). The percentages of variance indicated in the PCoA plots correspond to the variance explained by Axis 1 alone, which captures the highest proportion of compositional differences between the samples.

#### 3.3.2. Taxonomic Distribution at Phylum and Genus Levels

In samples ine17 to ine20, taxonomic classification at the phylum level revealed *Ascomycota* as the dominant phylum across all samples (Figure 8; Appendix A). However, control compost samples (ine17, ine18) showed a higher relative abundance of *Basidiomycota* compared to leather compost samples (ine19, ine20), which exhibited a greater proportion of unidentified phyla.

At the genus level, *Mycothermus* was the most abundant genus across both control and leather compost samples (Figure 9; Appendix A). Other genera showed differential presence across sample types. *Chrysosporium* was more frequently detected in control compost samples, while an unclassified genus from the family *Chaetomiaceae* appeared predominantly in leather compost samples.

Genera more abundant in leather compost samples included *Thermomyces* and *Remersonia*. In contrast, *Scutellinia* and *Thelebolus* were more frequently detected in control compost samples. In samples ine21 to ine26, *Ascomycota* was the predominant phylum, representing more than 95% of relative abundance in all cases (Figure 10; Appendix A).

At the genus level, notable differences were observed between sample types. *Mycothermus* was dominant in control compost, whereas *Melanocarpus* was predominant in leather compost samples (Figure 11; Appendix A). The control sample corresponds to the initial time point of the composting process, while the leather compost samples represent the final state. Additional genus-level assignments are available in the Appendix A. *Melanocarpus* and *Thermomyces* were primarily detected in leather compost samples, while *Remersonia* and *Dipodascus* were more frequently found in the control group.

### 3.4. Cultivable Bacterial and Fungal Species Identification

After the 44-day composting period, samples were collected from both 2 L and 40 L composting vessels for microbial cultivation. A total of 30 bacterial strains and 1 fungal strain were isolated. Strains labelled with the prefix “C” were obtained from microbial biofilms formed directly on composted leather fragments, while those labelled “Comp” were isolated from the compost matrix surrounding the leather. Each isolate was identified by partial gene sequencing (*16S rRNA* for bacteria and *23S rRNA* for fungi), and the corresponding sequences were deposited in the NCBI GenBank database. The final validated taxonomic assignments and accession numbers are listed in Table 2 (bacteria) and Table 3 (fungi), with the associated dendrograms provided in the Appendix A. Overall, most of the cultivated strains corresponded to isolates recovered from leather fragments (prefix “C”), while a smaller subset (prefix “Comp”) was obtained from the compost matrix surrounding the leather.

### 3.5. Scanning Electron Microscopy (SEM) & Gravimetric Analysis: Leather Analysis

Gravimetric analysis in the 2 L system indicated a 38.94% mass loss of the leather after 44 days (Figure 12). In the 40 L setup, leather degradation was so advanced that no substantial fragments (>1 cm) were recovered, precluding gravimetric analysis and suggesting nearly complete degradation (~100%). This improvement is attributed to bioaugmentation with mature compost from the 2 L system.

SEM observations were performed on residual fragments (>1 cm) of degraded leather recovered from the 40 L setup composting process. As shown in Figure 13, the characteristic fibrous structure of leather was preserved, confirming that the fragments corresponded to wet blue leather.

More advanced signs of microbial colonisation and surface disruption are evident in Figure 14 and Figure 15. At higher magnifications (×3500 to ×7000), the leather matrix appears heavily colonised by bacterial cells, with widespread biofilms and the presence of Extracellular Polymeric Substances (EPS). The EPS is visible as amorphous material bridging fibre gaps and embedding microbial cells. Microbial morphologies consistent with cocci and bacilli are clearly visible, indicating a multispecies colonisation. In some areas, matrix detachment, surface irregularity, and fibre separation are observed, indicating progressive biodegradation.

SEM analysis of untreated leather samples before composting revealed well-organised collagen fibrils with preserved periodic ultrastructure and no evidence of microbial colonisation or surface disruption (Figure 16). These control images provide a morphological baseline to evaluate the extent of degradation and biofilm formation observed in the composted samples.

## 4. Discussion

A key outcome of this work is that thermophilic composting, under the conditions tested, generated sufficiently harsh physicochemical environments to induce marked structural alteration of chromium-tanned leather. Macroscopically, leather pieces became brittle and fragmented, and scanning electron microscopy revealed the loss of the original fibre organisation, along with clear signs of microbial colonisation and extensive biofilm formation on leather surfaces. The clear bacterial colonisation of these fragments and the presence of mature biofilms confirm the active involvement of microbial consortia in leather degradation [58]. In addition, the abundant extracellular polymeric substances (EPS) observed in these biofilms not only indicate biofilm maturity but also suggest a potential mechanism for chromium tolerance, as EPS have been described as efficient agents in heavy metal binding and flocculation [35]. In the pilot-scale assay, the combination of higher temperatures and longer thermophilic phases resulted in particularly extensive fragmentation, leaving only small remnants after 44 days; these fragments nevertheless provided suitable material for microscopic examination and microbiological analyses.

In this work, the two composting scales were conceived as complementary experimental tools rather than as alternative strategies to be compared in terms of performance. The 2 L vessels provided a convenient laboratory setup to run parallel assays with and without leather under closely matched conditions, facilitating the detection of leather-associated changes in compost microbial communities. In contrast, the 40 L reactor reproduced a more realistic composting scenario, reached higher peak temperatures, and promoted a more extensive physical alteration of the leather pieces, which enabled examination of colonisation patterns and biofilm formation by microscopy. Thus, the choice between small- and larger-scale systems will depend on whether the main objective is to perform controlled comparative assays or to approximate pilot-scale composting conditions, rather than on differences in degradation efficiency per se. In contrast to the small-scale assay, where leather-amended and control vessels were sampled at the same composting stage and differences can therefore be primarily linked to the presence of chromium-tanned leather, the pilot-scale reactor does not allow the effect of leather and composting time to be fully disentangled. The pronounced shifts in bacterial and fungal communities observed between day 0 and day 44 in the 40 L system are likely primarily driven by the progression of composting itself, including the thermophilic temperature regime and the transformation and depletion of organic substrates, with leather acting as an additional, but not exclusive, selective factor. Taken together, the samples analysed in this study provide a coherent, albeit non-factorial, picture of the microbiota associated with chromium-tanned leather under thermophilic composting. The 40 L sample at time zero represents the initial compost community before leather addition, while the 2 L control vessels sampled after 44 days represent a thermophilic compost community that has evolved in the absence of leather. The corresponding 2 L vessels with leather and the 40 L reactor at day 44 represent thermophilic compost communities that have matured in the presence of leather. This hybrid design allowed us to obtain a baseline for the starting compost, a reference for mature compost without leather and two complementary views of compost matured with leather. At the same time, we explicitly acknowledge that the effect of composting time and leather cannot be fully disentangled in the large-scale system.

At the community level, bacterial and fungal sequencing data confirmed that composting of leather induces a significant shift in microbial composition and diversity. Bacterial richness and diversity were markedly reduced in leather compost samples compared with controls, consistent with the selective pressure imposed by chromium and collagen-rich substrates. Nevertheless, several genera such as *Sinibacillus*, *Desulfotomaculum*, *Chelativorans*, and *Bdellovibrio* were enriched in the leather compost environment, many of which have been reported in composts containing organic nitrogen and/or metal-contaminated matrices [59,60,61,62].

Differences were observed between the genera present in the leather compost samples and those in the control compost. Among the genera detected exclusively in the leather-degraded samples were *Novibacillus*, *Aeribacillus*, and *Thermobifida*, all of which have been previously described as thermophilic organisms involved in degradation processes under high-temperature composting conditions [55,56,57]. The presence of these genera supports the adaptive capacity of microbial communities in thermophilic composting environments and their potential role in breaking down recalcitrant materials, such as collagen. In contrast, the genus *Ruminofilibacter* was detected only in control compost samples and was absent from those containing leather. This genus has been described as an efficient degrader of lignocellulose and has been isolated from anaerobic digesters fed with manure [54]. Its absence in the leather compost samples may indicate sensitivity to tanned waste or limited adaptation to the specific substrates in this type of compost. Notably, *Ruminofilibacter* was also identified in the 2 L control compost, further reinforcing its specificity to specific composting environments. The presence of *Actinobacteria* in leather compost samples is also noteworthy. Although not among the most abundant phyla, their detection aligns with their known roles in the degradation of collagen and other polymeric materials. Actinobacteria possess a broad enzymatic repertoire, including collagenases and hydrolases that degrade proteinaceous substrates and synthetic polymers such as polylactic acid and rubber [63,64,65]. Their ecological relevance in composting environments has been highlighted in previous studies, particularly within families such as *Streptomycetaceae* and *Pseudonocardiaceae*, which can produce enzymes that initiate polymer breakdown [66]. These findings support their potential contribution to the microbial consortia involved in leather biodegradation, especially under thermophilic conditions. The fungal community also exhibited notable shifts, with *Melanocarpus*, *Thermomyces*, and *Remersonia* being characteristic of leather compost samples. These genera are frequently described in thermophilic composting and are known producers of thermostable hydrolases [67]. Their presence correlates with the observed morphological damage in SEM imaging, where fibre disruption and the loss of the periodic collagen structure were evident.

Among the cultivable strains identified in this study, we observed a high prevalence of thermotolerant and stress-resistant genera such as *Bacillus* [68,69], *Paenibacillus*, *Aneurinibacillus*, *Ureibacillus*, *Aeribacillus* and *Chelatococcus*, many of which have been reported in high-temperature composting environments and are known for their keratinolytic or proteolytic potential. Several of the isolates recovered here have also been previously associated with biodegradation and bioremediation processes. For example, *Rhodococcus rhodochrous* has shown a high capacity to adsorb heavy metal ions such as cadmium (II), lead (II), nickel (II), cobalt (II) and hexavalent chromium (VI) through extracellular polymeric substances [70]; *Bacillus pumilus* and *Bacillus subtilis* have been described as bacteria capable of tolerating and helping to remove elevated concentrations of different metals [71,72]; and *Pseudomonas aeruginosa* and *Acinetobacter* spp. are well-known for their ability to remove or transform heavy metals, including chromium [73,74,75]. Other genera, such as *Ochrobactrum ciceri* and *Pseudoxanthomonas*, have been highlighted for their potential in metal bioremediation and lignocellulose degradation, respectively [76,77,78]. The isolation of *Thermomyces lanuginosus*, together with the detection of *Thermomyces*, *Melanocarpus* and *Thermobifida* in leather compost samples, further points to thermophilic fungi and actinobacteria as important sources of thermostable hydrolases [57,67,79,80]. Overall, the predominance of thermophilic and proteolytic genera in both the culture collection and the amplicon data supports the hypothesis that enzymatic hydrolysis of proteinaceous substrates, particularly collagen, is a major mechanism contributing to the observed breakdown of chromium-tanned leather. Although ITS amplicon sequencing revealed a diverse fungal community in the composting systems, only a single filamentous fungus was recovered in our culture collection. This reflects that the isolation step was intentionally focused on bacteria, using nutrient-rich media and incubation conditions tailored to fast-growing thermotolerant bacteria rather than to fungi. We acknowledge this as a limitation and as an opportunity for future work using dedicated protocols for fungal isolation. The consistent presence of biofilm-producing bacteria, such as *Bacillus* and *Paenibacillus*, supports the hypothesis that microbial adhesion and localised enzyme activity may be critical in initiating collagen disruption. These findings are consistent with previous reports highlighting the role of extracellular enzymes and acidic metabolites in leather biodegradation pathways. The observed shifts in microbial composition between control and leather compost reflect selective ecological pressures imposed by chromium and collagen-rich substrates. The reduced alpha diversity in the leather compost suggests that only a limited subset of resilient or specialised taxa can thrive under these conditions, potentially influencing the overall microbial ecology and enzymatic landscape of compost. This selection may alter nutrient cycling dynamics and microbial succession in composting environments enriched with tanned leather waste. SEM observations revealed progressive matrix deterioration in leather samples, consistent with microbial enzymatic action. Images showed characteristic detachment of fibre bundles, amorphous regions lacking structural order, and extensive biofilm coverage. These results corroborate previous findings that the mechanical integrity of collagen can be disrupted by sustained microbial activity, particularly in the presence of extracellular enzymes and acidic metabolic byproducts. Gravimetric analysis, although limited to the small-scale system due to complete degradation in the large-scale setup, provides further support for microbial degradation. The variable mass loss among samples suggests heterogeneous microbial dynamics, potentially influenced by initial colonisation, leather accessibility, and local physicochemical gradients within the compost matrix. While it is acknowledged that thermal and chemical factors may contribute to the overall degradation of leather during composting, the results of this study point towards microbial activity as the primary driver. Thermophilic composting systems inherently generate elevated temperatures that can facilitate the breakdown of organic materials, but the degradation of collagen-rich substrates such as chromium-tanned leather is unlikely to be solely explained by abiotic processes. The complete disintegration observed in the large-scale system, together with the dense microbial colonisation detected by SEM imaging and the taxonomic profiling of known proteolytic and thermotolerant genera, strongly supports a biologically driven mechanism. Moreover, the identification of specific bacterial and fungal taxa with documented enzymatic potential for protein and polymer degradation reinforces the conclusion that microbial consortia played a central role in the observed breakdown of leather. While heat and chemical reactions may have accelerated the process or influenced bioavailability, they do not account for the structural alterations, biofilm formation, and microbial selectivity documented in this study. This study helps address the current knowledge gap regarding the compostability of chromium-tanned leather under thermophilic conditions. Despite the relevance of composting as a potential valorisation route for leather waste, there is still limited understanding of how this material behaves within the composting matrix and how microbial consortia interact with its chemically stabilised collagen structure. Our results provide the first integrated insight into the taxonomic composition, colonisation patterns, and structural degradation associated with leather biodegradation in composting conditions. In particular, the SEM observations confirm extensive microbial colonisation and biofilm formation on leather fragments, supporting the role of biological activity in the degradation process. The metagenomic profiling further revealed bacterial and fungal genera previously unreported in leather degradation studies, highlighting novel microbial candidates potentially involved in collagen decomposition and metal tolerance.

Future research should aim to characterise the metabolic activity and enzymatic pathways of these key microbial taxa through metaproteomics or transcriptomics. In addition, analysing the chemical composition of the final compost, with particular attention to chromium speciation and possible oxidation states, will be crucial to assess the safety and environmental impact of the process. Understanding whether chromium is mobilised, transformed, or stabilised during composting is essential to determine the quality and suitability of the end product for agricultural or environmental use.

## 5. Conclusions

This study confirms that thermophilic composting environments can support the microbial degradation of chromium-tanned leather (wet blue) and facilitate the identification of microorganisms adapted to such conditions. By establishing controlled composting systems, we isolated and characterised microbial strains from both compost and leather surfaces. Among the most notable findings are the presence of thermotolerant bacterial genera such as *Sinibacillus*, *Chelativorans*, and *Bdellovibrio*, as well as fungal genera such as *Thermomyces*, all associated with the degradation of complex proteinaceous materials. The isolation of 30 bacterial and 1 fungal cultivable strains, many of which exhibit metal tolerance and proteolytic capabilities, further supports the potential of compost-derived microbiota for future biotechnological applications. Overall, the study helps fill a knowledge gap regarding microbial consortia active in leather degradation under chromium stress and provides a solid foundation for the development of targeted biotechnological solutions aligned with circular economy goals.

## Figures and Tables

**Figure 1 biology-14-01799-f001:**
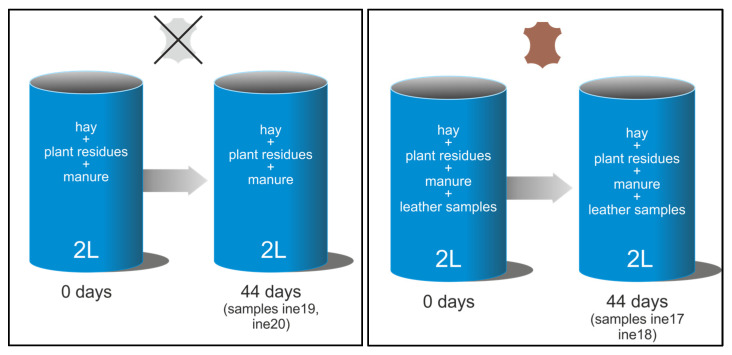
Experimental design of the small-scale (2 L) thermophilic composting system. Two 2 L Dewar vessels were prepared with the same compost matrix (hay + plant residues + manure). The no-leather vessel (**left**) contained only the compost matrix and was sampled after 44 days of thermophilic composting (samples ine19 and ine20). The leather vessel (**right**) contained the same compost matrix plus chromium-tanned leather strips and was also sampled after 44 days (samples ine17 and ine18). These samples were used for 16S rRNA and ITS amplicon sequencing and for strain isolation.

**Figure 2 biology-14-01799-f002:**
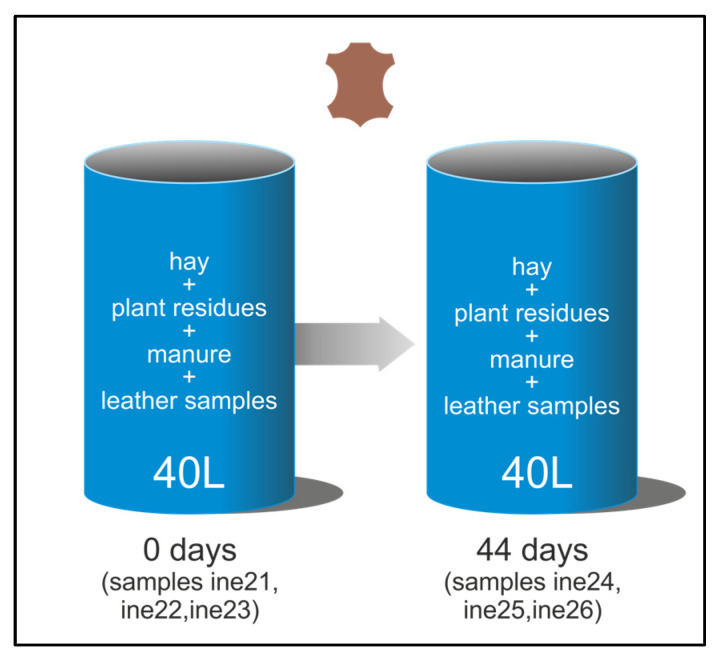
Experimental design of the pilot-scale (40 L) thermophilic composting system. A 40 L Dewar vessel was filled with the compost matrix (hay + plant residues + manure) and sampled at time zero, before leather addition, to characterise the initial compost community (samples ine21, ine22 and ine23). Chromium-tanned leather pieces were then added to the reactor, which was allowed to compost under thermophilic conditions for 44 days. At the end of the assay, compost and remaining leather fragments were sampled (samples ine24, ine25 and ine26) for 16S rRNA and ITS amplicon sequencing and for strain isolation.

**Figure 3 biology-14-01799-f003:**
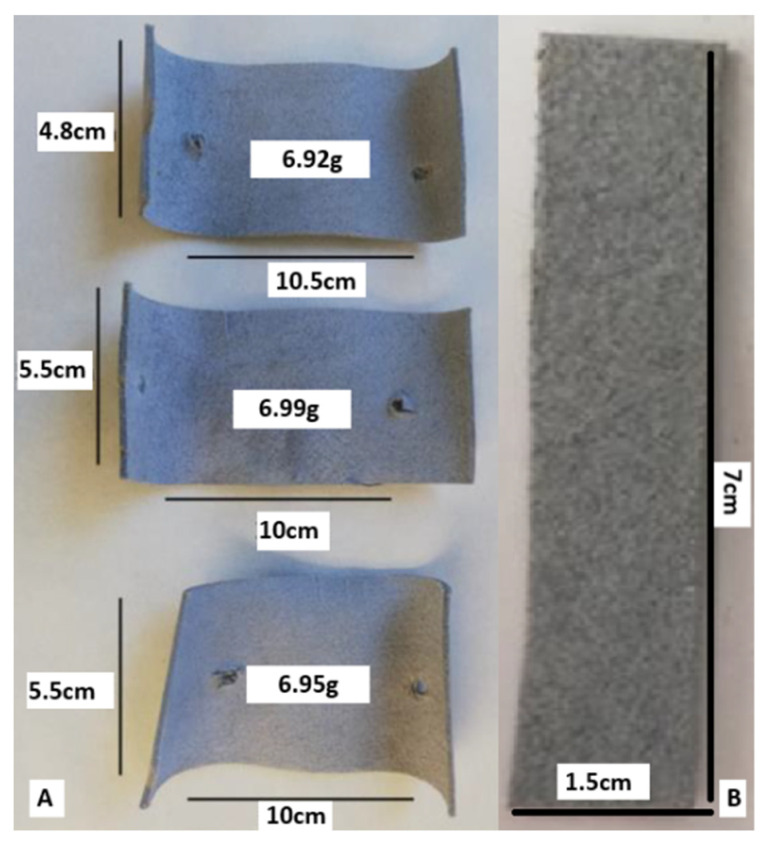
Wet blue leather samples used in composting assays. (**A**) Three rectangular leather samples (approx. 10 × 5.5 cm) placed in the 40 L Dewar vessel for large-scale composting, with initial weights measured before exposure. (**B**) One of the three leather strips (7 × 1.5 cm) used in the small-scale composting system (2 L Dewar vessel), fixed vertically on metal rods to ensure stability within the compost matrix.

**Figure 4 biology-14-01799-f004:**
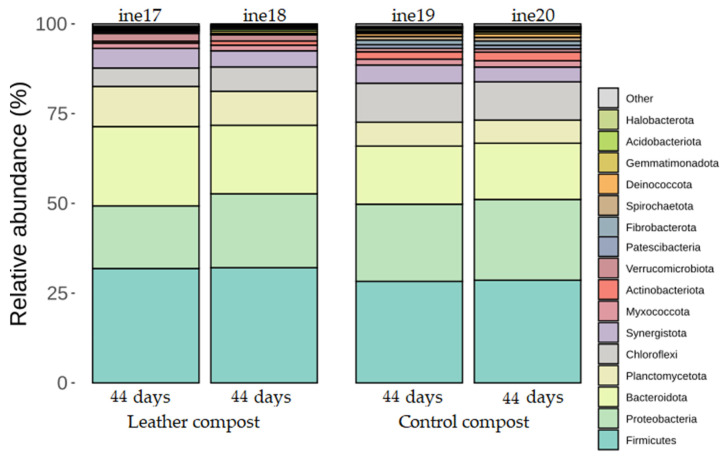
Taxonomic distribution at the phylum level for samples ine17–ine20 (small-scale 2 L Dewar vessel). All samples shown were analysed for *16S rRNA* gene-based taxonomic profiling. The “Others” category includes bacterial taxa with annotations that do not reach the phylum level and/or exhibit low relative abundance across samples.

**Figure 5 biology-14-01799-f005:**
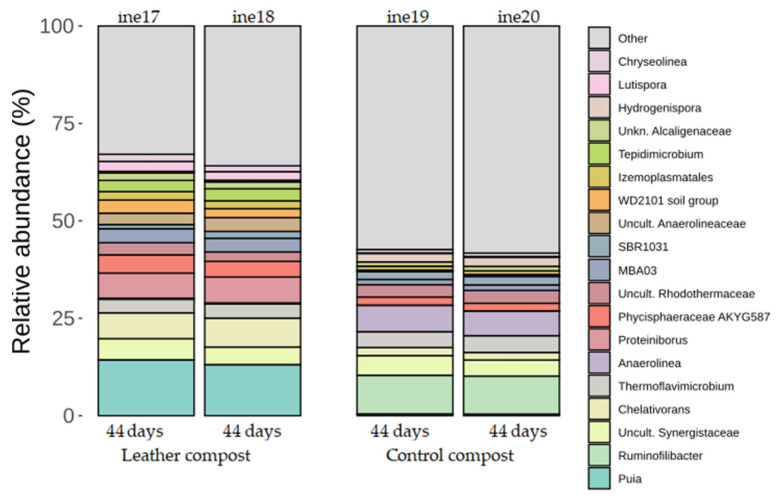
Taxonomic distribution at the genus level for samples ine17–ine20 (small-scale 2 L Dewar vessel). All samples shown were analysed for *16S rRNA* gene-based taxonomic profiling. The “Others” category includes bacterial taxa with annotations that do not reach the phylum level and/or exhibit low relative abundance across samples.

**Figure 6 biology-14-01799-f006:**
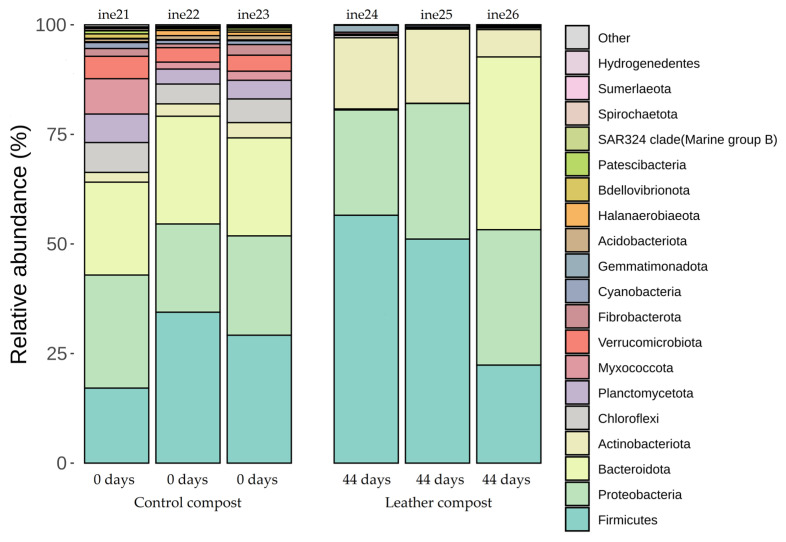
Taxonomic distribution at the phylum level for samples ine21–ine26 (large-scale 40 L Dewar vessel). All samples shown were analysed for *16S rRNA* gene-based taxonomic profiling. The “Others” category includes bacterial taxa with annotations that do not reach the phylum level and/or exhibit low relative abundance across samples.

**Figure 7 biology-14-01799-f007:**
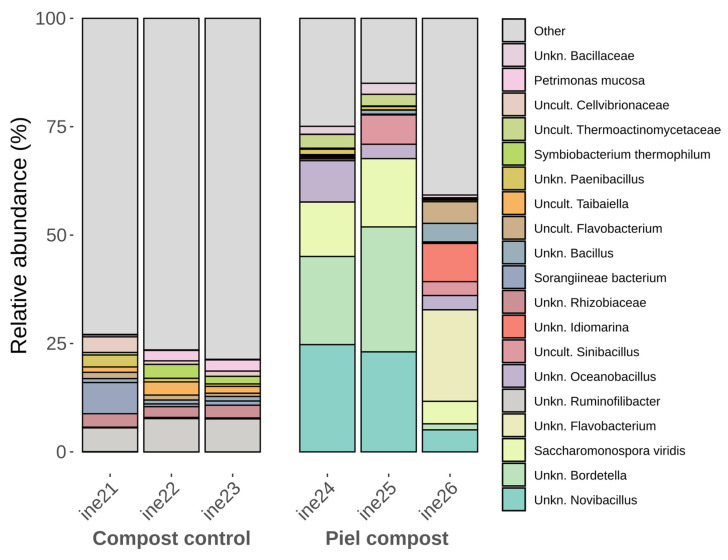
Taxonomic distribution at the genus level for samples ine21–ine26 (large-scale 40 L Dewar vessel). All samples shown were analysed for *16S rRNA* gene-based taxonomic profiling. The “Others” category includes bacterial taxa with annotations that do not reach the phylum level and/or exhibit low relative abundance across samples.

**Figure 8 biology-14-01799-f008:**
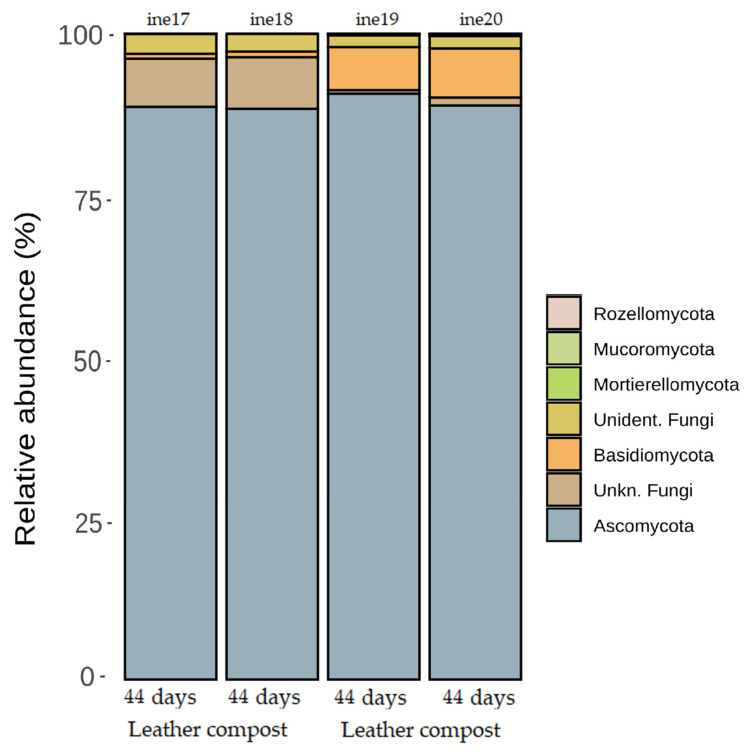
Taxonomic distribution at the phylum level for samples ine17–ine20 (small-scale 2 L Dewar vessel). All samples shown were analysed for taxonomic profiling based on the *23S rRNA* gene.

**Figure 9 biology-14-01799-f009:**
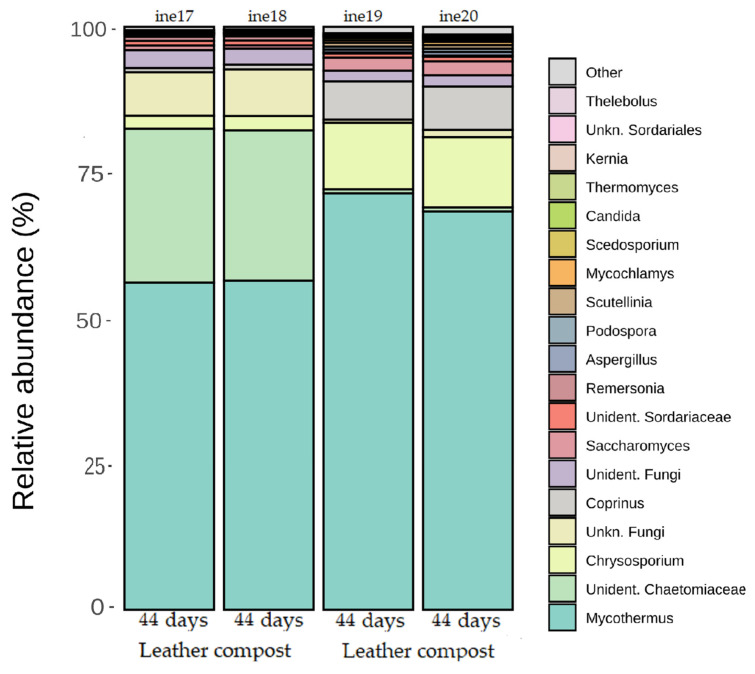
Taxonomic distribution at the genus level for fungal samples ine17–ine20 (small-scale 2 L Dewar vessel). All samples shown were analysed for taxonomic profiling based on the *23S rRNA* gene. The “Others” category includes fungal taxa with annotations that do not reach the phylum level and/or exhibit low relative abundance across samples.

**Figure 10 biology-14-01799-f010:**
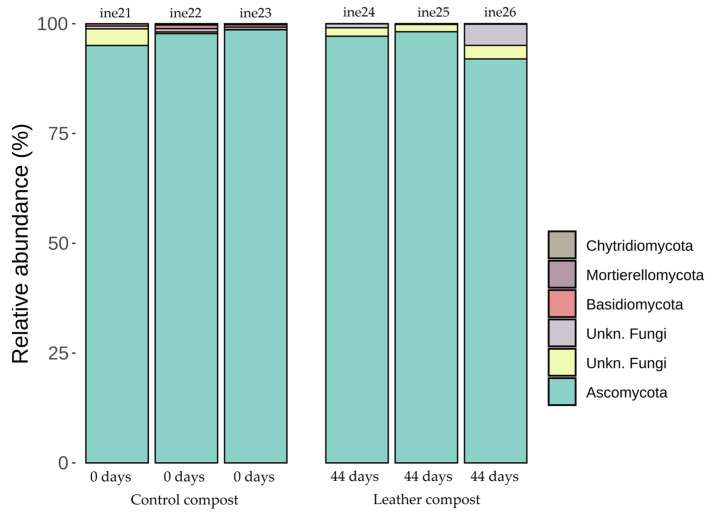
Taxonomic distribution at the phylum level for fungal samples ine21–ine26 (large-scale 40 L Dewar vessel). All samples shown were analysed for taxonomic profiling based on the *23S rRNA* gene.

**Figure 11 biology-14-01799-f011:**
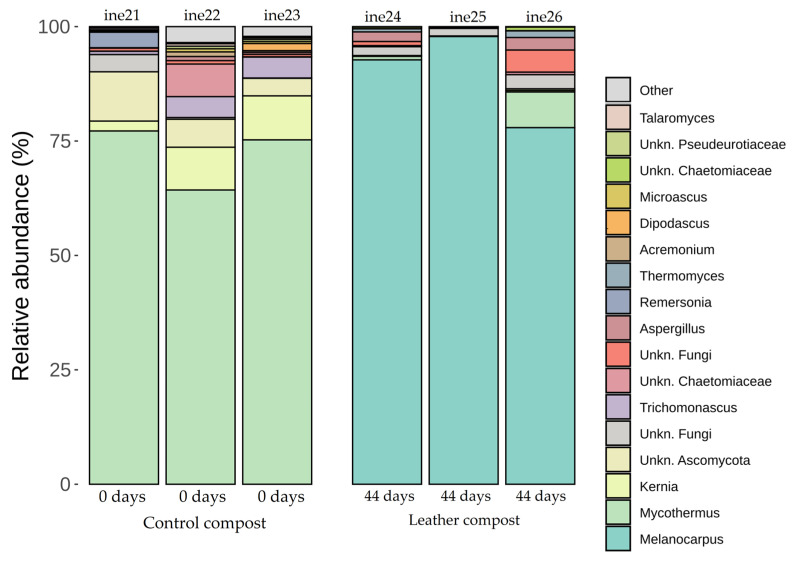
Taxonomic distribution at the genus level for fungal samples ine21–ine26 (large-scale 40 L Dewar vessel). All samples shown were analysed for taxonomic profiling based on the *23S rRNA* gene. The “Others” category includes fungal taxa with annotations that do not reach the phylum level and/or exhibit low relative abundance across samples.

**Figure 12 biology-14-01799-f012:**
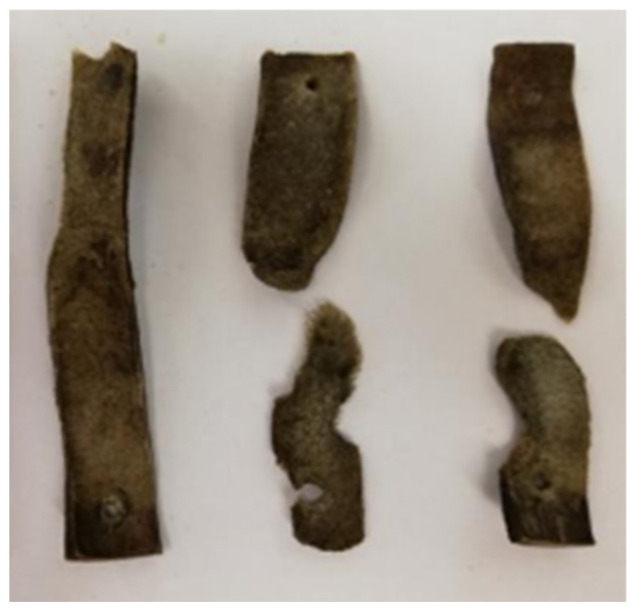
The three chromium-tanned leather strips (7 × 1.5 cm) recovered after 44 days of composting in the small-scale system (2 L Dewar vessel). The strips were fixed initially vertically on metal rods within the compost matrix; after composting, the second and third strips (from left to right) appear split into two pieces.

**Figure 13 biology-14-01799-f013:**
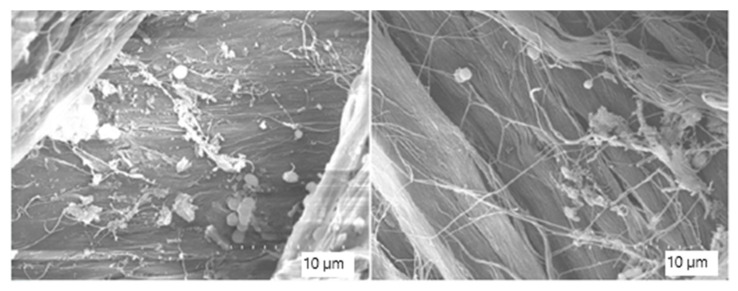
Scanning electron microscopy (SEM) images of composted leather fragments. Characteristic collagen fibres of wet blue leather are visible. Images captured at ×2500 magnification.

**Figure 14 biology-14-01799-f014:**
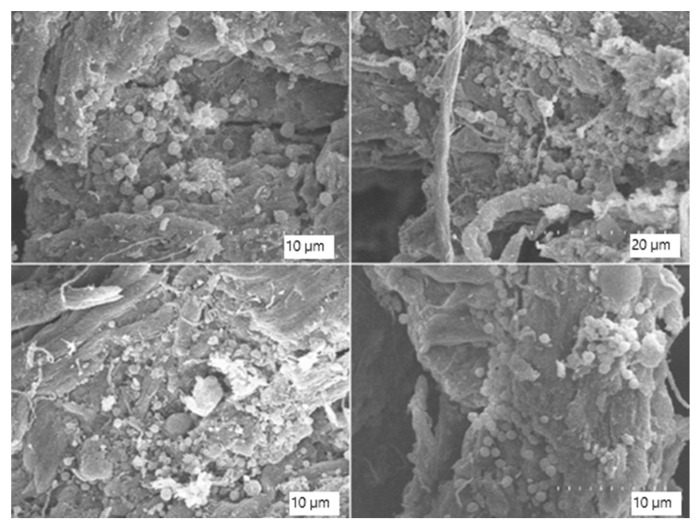
Scanning electron microscope (SEM) images of the composted leather surface showing extensive bacterial colonisation and biofilm formation. Images captured at ×3500 magnification.

**Figure 15 biology-14-01799-f015:**
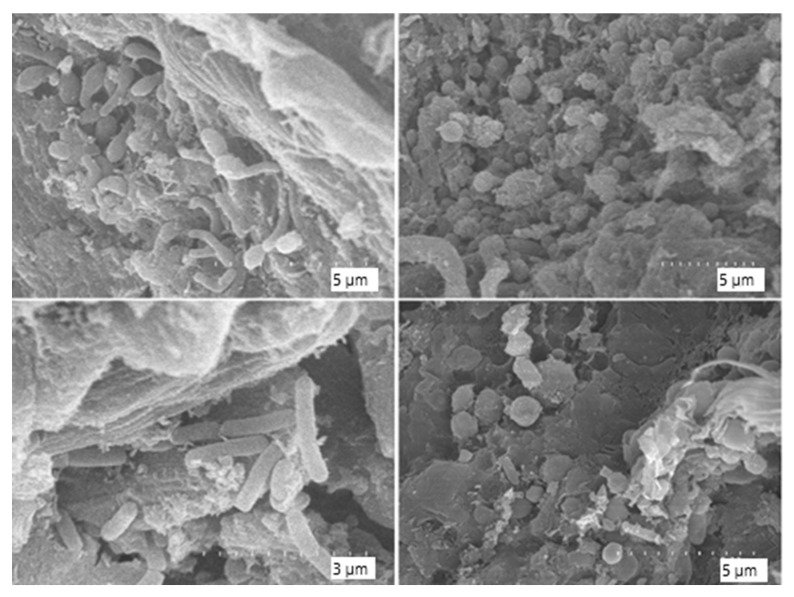
Scanning electron microscope (SEM) images of composted leather surface at higher magnification (×7000), revealing dense bacterial coverage and abundant extracellular matrix consistent with mature biofilms.

**Figure 16 biology-14-01799-f016:**
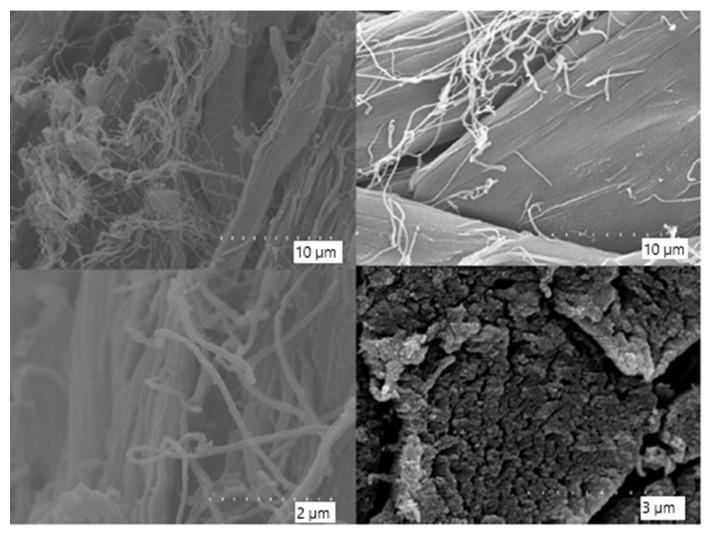
Scanning Electron Microscopy (SEM) images of the negative control leather sample before composting, showing the characteristic ultrastructure of untreated collagen. The periodic arrangement and compact organisation of collagen fibrils is preserved, with no visible microbial colonisation or surface damage. Images were captured at magnifications ranging from ×10,000 to ×50,000.

**Table 1 biology-14-01799-t001:** Description of compost samples analysed for DNA extraction and metagenomic sequencing. Samples were collected from two composting systems: a small-scale 2 L Dewar vessel (ine17–ine20) and a large-scale 40 L Dewar vessel (ine21–ine26). Each sample is classified by origin (composted leather or control compost) and by composting duration (0 or 44 days). Sample numbering starts at 17 to maintain continuity with previous related experiments.

Sample Name	Vessel	Type	Time (Days)
ine17	2 L	Composted leather	44
ine18	2 L	Composted leather	44
ine19	2 L	Control compost	44
ine20	2 L	Control compost	44
ine21	40 L	Control compost	0
ine22	40 L	Control compost	0
ine23	40 L	Control compost	0
ine24	40 L	Composted leather	44
ine25	40 L	Composted leather	44
ine26	40 L	Composted leather	44

**Table 2 biology-14-01799-t002:** Identification of cultivable bacterial strains isolated from composted leather samples based on partial *16S rRNA* gene sequencing and BLAST analysis. Growth medium and incubation temperature used for isolation and maintenance are indicated for each strain. Strains labelled as “C” were isolated from biofilms directly on leather fragments, while “Comp” strains were isolated from the surrounding compost. NCBI GenBank accession numbers of the isolates are indicated.

Strain Code	Closest Species	Similarity	Temp.	Media	GeneBank
C4	*Rhodococcus rhodochrous*	98.39%	30 °C	R2A	PX490265
C5	*Brevundimonas naejangsanensis*	99.43%	30 °C	R2A	PX490266
C6	*Bacillus zhangzhouensis*	99.15%	30 °C	R2A	PX490267
C7	*Bacillus pumilus*	99.17%	30 °C	R2A	PX490268
C8	*Pseudomonas aeruginosa*	99.84%	30 °C	R2A	PX490269
C9	*Burkholderia arboris*	98.81%	30 °C	R2A	PX490270
C10	*Empedobacter brevis*	99.24%	30 °C	TSA	PX490271
C11	*Bacillus subtilis*	100%	30 °C	TSA	PX490272
C12	*Acinetobacter beijerinckii*	98.96%	30 °C	TSA	PX490273
C13	*Glutamicibacter mishrai*	99.52%	30 °C	TSA	PX490274
C14	*Bacillus velezensis*	99.17%	30 °C	TSA	PX490275
C15	*Bacillus smithii*	99.34%	55 °C	YM	PX490276
C16	*Klebsiella variicola*	98.74%	55 °C	YM	PX490277
C17	*Aneurinibacillus thermoaerophilus*	98.43%	55 °C	YM	PX490278
C18	*Burkholderia arboris*	99.44%	55 °C	TSA	PX490279
C19	*Chelatococcus composti*	99.45%	55 °C	TSA	PX490280
C20	*Bacillus paramycoides*	99.78%	55 °C	GYM	PX490281
C21	*Ralstonia pickettii*	98.96%	55 °C	GYM	PX490282
C22	*Pantoea cypripedii*	97.04%	55 °C	GYM	PX490283
Comp414.1	*Brucella ciceri*	98.28%	30 °C	R2A	PX490284
Comp414.2	*Chryseobacterium jejuense*	99.17%	30 °C	R2A	PX490285
Comp414.3	*Brevibacillus parabrevis*	95.65%	30 °C	R2A	PX490286
Comp419	*Bacillus haynesii*	98.80%	55 °C	TSA	PX490287
Comp424	*Aeribacillus composti*	99.23%	55 °C	NAI	PX490288
Comp438	*Bacillus licheniformis*	95.25%	55 °C	YM	PX490289
Comp440	*Heyndrickxia coagulans*	98.25%	55 °C	YM	PX490290
Comp442	*Ureibacillus thermosphaericus*	99.63%	55 °C	TSA	PX490291
Comp443	*Aeribacillus composti*	99.41%	55 °C	TSA	PX490292
Comp445	*Bacillus haynesii*	97.99%	55 °C	NAI	PX490293

**Table 3 biology-14-01799-t003:** Identification of cultivable fungal strains isolated from composted leather samples based on partial ITS gene sequencing and BLAST analysis. Growth medium and incubation temperature used for isolation and maintenance are indicated for each strain. The isolate marked as “Comp” was obtained from the compost material surrounding the leather. GenBank accession numbers of the isolate.

Strain Code	Closest Species	Similarity	Temp.	Media	GeneBank
Comp401	*Thermomyces lanuginosus*	99.64%	100 °C	YM	PX490265

## Data Availability

The datasets generated and analysed during the current study are available from the corresponding author upon reasonable request.

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
