# Peer review of "Microbial Degradation of Chromium-Tanned Leather During Thermophilic Composting: A Multi-Scale Analysis of Microbial Communities and Structural Disruption"

_biology, 2025, doi:10.3390/biology14121799_

Round 1
Reviewer 1 Report
Comments and Suggestions for Authors
The authors in this manuscript demonstrated the microbial degradation of Chromium-tanned leather through thermophilic composting. The results provide the taxonomic composition, colonization patterns of microorganisms, and structural degradation associated with leather biodegradation in composting conditions. Although this study is well designed, scientifically sound, and impressive, some concerns need to be addressed.
Comments:
- In this study, a total of 30 bacterial strains and 1 fungal strain were isolated. Why was only 1 fungal strain able to be isolated, despite a large fungal diversity being detected in the composting samples? You may justify it in the discussion section.
- In Figure 10, Leather pieces shown in Figure 13B were composted in a 2 L vessel for 44 days. Here, please clarify the mention of “Figure 13B”.
- Composting experiments were conducted in both small-scale (2 L) and large-scale (40 L Dewar vessels) setups. Is there any notable difference in their results? Which one is a comparatively more effective or recommendable strategy that can be adopted by someone, and why? You may state it in the discussion or conclusion section.
- In the “Simple summary” section, it is mentioned that over 40 days, the leather was 20 significantly broken down. Is it 40 days or 44 days? Please check.
Author Response
The authors in this manuscript demonstrated the microbial degradation of Chromium-tanned leather through thermophilic composting. The results provide the taxonomic composition, colonization patterns of microorganisms, and structural degradation associated with leather biodegradation in composting conditions. Although this study is well designed, scientifically sound, and impressive, some concerns need to be addressed.
Comments:
- In this study, a total of 30 bacterial strains and 1 fungal strain were isolated. Why was only 1 fungal strain able to be isolated, despite a large fungal diversity being detected in the composting samples? You may justify it in the discussion section. In this study, the cultivation step was deliberately focused on bacteria, because the next stages of our research were planned to use bacterial consortia in standardised leather biodegradation assays. For this reason, we used nutrient-rich media and incubation conditions tailored to fast-growing thermotolerant bacteria, without specifically optimising fungal isolation. Under these conditions, only one filamentous fungus was consistently recovered as a pure culture, and we decided to include it in the collection due to its association with compost in contact with leather. We agree that this under-represents the fungal diversity observed by ITS sequencing, and we now acknowledge this limitation in the Discussion.
- In Figure 10, Leather pieces shown in Figure 13B were composted in a 2 L vessel for 44 days. Here, please clarify the mention of “Figure 13B”. The mention was a typographical error and has been updated.
- Composting experiments were conducted in both small-scale (2 L) and large-scale (40 L Dewar vessels) setups. Is there any notable difference in their results? Which one is a comparatively more effective or recommendable strategy that can be adopted by someone, and why? You may state it in the discussion or conclusion section. We thank the reviewer for this observation. In our study, the small- and large-scale composting systems were not designed to be compared as alternative strategies for optimising leather degradation, but to provide complementary experimental contexts for microbiological characterisation. The 2 L vessels enabled parallel assays with and without leather under closely matched conditions, helping identify leather-associated shifts in the compost microbiota. The 40 L reactor was used to reproduce a more realistic composting scenario, to follow the temporal evolution of the system and to obtain larger pieces of leather for microscopic examination of colonisation and biofilm formation. Although the larger reactor reached higher peak temperatures and led to greater physical disintegration of the leather pieces, we did not aim to quantify which scale was “more effective” for degradation. Instead, we view both setups as complementary tools that can be selected based on the objectives of a given study (e.g., controlled comparative assays versus more realistic pilot-scale composting).
- In the “Simple summary” section, it is mentioned that over 40 days, the leather was 20 significantly broken down. Is it 40 days or 44 days? Please check. In the revised version of the manuscript, the Simple Summary has been completely rewritten.
Reviewer 2 Report
Comments and Suggestions for Authors
The manuscript entitled:"Microbial Degradation of Chromium-Tanned Leather During Thermophilic Composting: A Multi-Scale Analysis of Community Dynamics and Structural Disruption" was submitted to Biology.
The purpose of this research is to provide readers with a practical application for the safe disposal of industrial waste resistant to natural breakdown. This involves using the composting media to activate microorganisms under effective conditions, thereby breaking down waste: “Chromium-Tanned Leather”, which is highly toxic to the environment. Controlling the optimal conditions for this biological decomposition process is a critical and challenging task.
Therefore, I commend that the research has covered many aspects of maintaining optimal conditions, whether using 2 liters or 40 liters of compost for the growth of different types of thermophilic microbes that are active in the biodegradation of Chromium-Tanned Leather.
Few comments find in the attached manuscript to support research purpose, so please answer carefully

Author Response
The manuscript entitled:"Microbial Degradation of Chromium-Tanned Leather During Thermophilic Composting: A Multi-Scale Analysis of Community Dynamics and Structural Disruption" was submitted to Biology.
The purpose of this research is to provide readers with a practical application for the safe disposal of industrial waste resistant to natural breakdown. This involves using the composting media to activate microorganisms under effective conditions, thereby breaking down waste: “Chromium-Tanned Leather”, which is highly toxic to the environment. Controlling the optimal conditions for this biological decomposition process is a critical and challenging task.
Therefore, I commend that the research has covered many aspects of maintaining optimal conditions, whether using 2 liters or 40 liters of compost for the growth of different types of thermophilic microbes that are active in the biodegradation of Chromium-Tanned Leather.
Few comments find in the attached manuscript to support research purpose, so please answer carefully
We sincerely thank the reviewer for their positive assessment and for highlighting the potential of our work for the safe disposal of chromium-tanned leather. We fully share the long-term aim of advancing biological strategies for managing tannery waste. In fact, this study forms part of the core research of the first author’s PhD project, in which we ultimately aim to identify and develop microbial and enzymatic tools that may support safer treatment and valorisation of chromium-tanned leather residues. At present, compost derived from chromium-tanned leather is not considered acceptable for direct agronomic use because chromium largely remains in the solid fraction. This is precisely why it is essential to identify microbial communities and strains with potential for bioremediation or for assisting in future biological treatment strategies.
However, we wish to clarify that the specific aim of this manuscript is more fundamental and does not yet offer a ready-to-use practical protocol for safe waste disposal. Our primary goal is to establish a controlled thermophilic composting system and use it as an experimental environment to (i) characterise the microbial communities associated with chromium-tanned leather under thermophilic conditions, (ii) analyse colonisation and biofilm formation on leather surfaces, and (iii) isolate representative cultivable strains for further biotechnological studies. The details we provide on process conditions and the two composting scales are intended to document the robustness and suitability of the system for microbiological characterisation, rather than to compare or optimise alternative disposal strategies.
To avoid any misunderstanding, we have revised the title, Simple Summary, Abstract, Introduction and Discussion to explicitly state that this work does not aim to validate thermophilic composting as an immediately applicable safe disposal method, but rather to generate microbiological knowledge and strain collections that can underpin future development of biological technologies for chromium-tanned leather waste treatment. We hope that this clarification aligns the manuscript more clearly with both the reviewer’s expectations and our intended research trajectory.
Why authors did not draw a graphical abstract for this design, from reviewer suppose, it'll give more strength and clarification for the aim of this article. We agree that a visual summary of the experimental design can help clarify the aim and structure of the study. In the revised manuscript, we have therefore added a schematic overview (figure 1 and 2) of the composting setups and sampling strategy, which illustrates the two composting systems, the presence/absence of leather, and the main sampling points.
What is meant by ine.. clarify. The prefix “INE” in the strain codes is simply an internal identifier used in the strain collection of INESCOP (the research institute where the work was carried out) and does not carry any additional experimental meaning. To avoid confusion, we have now clarified this in the text.
Please put this reference. Done.
Authors should specify the temperature used for these media cause these isolates have specific temperature and preservative conditions for cultivation. We agree that specifying the incubation temperature of the media is important, given the thermophilic nature of the isolates. In the revised version of the manuscript, we now indicate the incubation temperature used for the cultivation of each group of isolates in the text and in the footnotes of the tables reporting the strains.
Reviewer 3 Report
Comments and Suggestions for Authors
Dear authors,
Despite the trend of increasing the environmental friendliness of industrial production, optimal methods of their disposal have not yet been developed for many types of industrial waste. This also applies to leathers treated with substances that inhibit the growth of microorganisms. Therefore, it makes sense to study different ways of their disposal using modern scientific approaches. Therefore, the goal of the work set by the authors seems interesting to me.
The introduction of the manuscript well describes the current state of the issue under study. The research methods are mostly described in detail. The acquisition, processing, and analysis of gene and metagenomic data correspond to the average level of articles in this field. The design of the experiments is acceptable. At the current stage, most of the metagenomic studies are limited to a small number of analyzed samples. Due to this limitation, I would prefer to analyze 10-12 samples from one full-scale experiment than 4-6 samples from two different ones. This is my personal opinion. The text of the manuscript is structured and written in a good style. The illustrations are relevant, informative and well executed. The conclusions of the work follow from the results obtained.
Questions and comments
- The phrase Community Dynamics in the title of the article can only be attributed to one of the experimenters (40 L, 0 and 40 day). Maybe the word dynamics should be removed from the title?
- It was incorrect to describe and discuss experiments in 2L and 40L in the same way. They differ in design. The differences in metagenomes in the 2L experiment can be explained by the addition of leather residues to the compost, since the control and experimental vessels were at the same stage of composting. In the 40L experiment, the differences in metagenomes can be explained by the action of two factors. The first is the addition of leather. The second was composting, which lasted 40 days, during which an increase in temperature and the transformation of the nutrient substrate changed the microbial community. The second factor seems to me stronger than the first one. This can explain the more significant shifts in Fig. 4-5 than in Fig. 2-3. The above is not adequately taken into account in the discussion and conclusions.
- The materials and methods describe the compost composition too briefly. What kind of hay, manure and plant residues was used? What was the nitrogen content?
- Why were the cultivated strains of microorganisms isolated only on rich media? Why wasn't a selective medium used to understand the presence of keratinolytic strains in the cultivated microbiota? The same applies to chromium resistance. This would make it possible to more confidently associate the isolated strains with destruction of chromium-tanned leather. Since many isolated strains were identified as bacilli, they could wait out unfavorable conditions in the compost in the form of spores, be metabolically inactive and not contribute to destruction of leather. After being exposed to nutrient media, the bacilli could restore metabolic activity and form colonies.
- In general, little attention has been paid to the description and discussion of cultivated strains. Were 2L and 40L vessels equally good at isolating leather-destroying microorganisms? From which vessels were more strains isolated? Such practical questions are important for those who will conduct similar research.
Short remark:
In Table 2, Table 3 the numbers of the strains from the GenBank closest to the isolated strains were not indicated. Although this was written in the description to the tables.
Author Response
Despite the trend of increasing the environmental friendliness of industrial production, optimal methods of their disposal have not yet been developed for many types of industrial waste. This also applies to leathers treated with substances that inhibit the growth of microorganisms. Therefore, it makes sense to study different ways of their disposal using modern scientific approaches. Therefore, the goal of the work set by the authors seems interesting to me.
The introduction of the manuscript well describes the current state of the issue under study. The research methods are mostly described in detail. The acquisition, processing, and analysis of gene and metagenomic data correspond to the average level of articles in this field. The design of the experiments is acceptable. At the current stage, most of the metagenomic studies are limited to a small number of analyzed samples. Due to this limitation, I would prefer to analyze 10-12 samples from one full-scale experiment than 4-6 samples from two different ones. This is my personal opinion. The text of the manuscript is structured and written in a good style. The illustrations are relevant, informative and well executed. The conclusions of the work follow from the results obtained.
Questions and comments
- The phrase Community Dynamics in the title of the article can only be attributed to one of the experimenters (40 L, 0 and 40 day). Maybe the word dynamics should be removed from the title? Our initial use of the term “Community Dynamics” in the title was intended to reflect the changes observed in microbial community composition under different conditions, namely the presence or absence of leather in the 2 L assays and the temporal evolution of the 40 L reactor (day 0 vs. day 44). In that sense, we did capture shifts in community structure associated with leather addition and the progression of composting. However, we agree that the experimental design, with only two sampling points in the 40 L system and a limited overall temporal resolution, could lead readers to expect a more detailed time-series analysis than we actually performed. To avoid overstating this aspect and better align the title with the study's scope, we have revised the title and removed the word “dynamics”.
- It was incorrect to describe and discuss experiments in 2L and 40L in the same way. They differ in design. The differences in metagenomes in the 2L experiment can be explained by the addition of leather residues to the compost, since the control and experimental vessels were at the same stage of composting. In the 40L experiment, the differences in metagenomes can be explained by the action of two factors. The first is the addition of leather. The second was composting, which lasted 40 days, during which an increase in temperature and the transformation of the nutrient substrate changed the microbial community. The second factor seems to me stronger than the first one. This can explain the more significant shifts in Fig. 4-5 than in Fig. 2-3. The above is not adequately taken into account in the discussion and conclusions. We thank the reviewer for this very relevant comment and fully agree that the 2 L and 40 L experiments differ in design and should not be interpreted similarly. In the 2 L assay, control and leather-amended vessels were sampled after 44 days of composting, so the observed differences in metagenomic profiles can primarily be attributed to the presence of chromium-tanned leather at this stage. In the 40 L reactor, no dedicated control vessel without leather was run in parallel. Still, compost samples were collected at time zero, before leather addition, and used as a baseline reference for microbial composition. The comparison between day 0 and day 44 in the 40 L system, therefore, reflects the combined effects of compost maturation and the presence of leather. As the reviewer correctly points out, the progression of composting (changes in temperature regime and substrate availability) is likely to be a stronger driver of the pronounced community shifts observed in Figures 4 and 5 than the leather itself. Taken together, the sampling scheme across both experiments provides information on (i) the initial compost community before leather addition (40 L, day 0), (ii) a thermophilic compost community after 44 days in the absence of leather (2 L control vessels) and (iii) thermophilic compost communities after 44 days in the presence of leather (2 L and 40 L assays). However, we agree that the large-scale experiment does not allow the isolated effect of leather to be formally separated from the impact of composting time, and we have revised the Discussion and Conclusions accordingly. We now explicitly state that the 2 L system is more appropriate for detecting leather-associated shifts under controlled conditions. In contrast, the 40 L system reflects the combined effect of compost maturation in the presence of leather. We also present both setups as complementary tools rather than directly comparable strategies.
- The materials and methods describe the compost composition too briefly. What kind of hay, manure and plant residues was used? What was the nitrogen content? In the revised version of the manuscript (Section 2.1), we now provide a more detailed description of the materials used. Specifically, both composting systems employed a mixture of cereal straw hay, green plant residues (mainly pruning waste from horticultural and ornamental plants) and mature livestock manure. The proportions used (hay 40%, plant residues 30%, and manure 30% in the 2 L vessels; hay 40%, manure 20%, and 10% mature compost in the 40 L reactor) were chosen to achieve a balanced carbon-to-nitrogen ratio in the initial mixture. Rather than targeting a specific total nitrogen percentage, the formulation was designed to fall within the C/N range (approximately 20–30) that is generally recommended for effective manure-based composting systems. This range is consistent with typical nitrogen contents and C/N ratios reported for cow and other livestock manures, as well as for hay and straw in the composting literature. We did not perform a separate elemental analysis of total nitrogen for the specific compost batch used in this experiment, and we now state this explicitly in the Materials and Methods. However, all treatments within each composting system used the same compost matrix, so comparisons between compost with and without leather are not affected by nitrogen content variability.
- Why were the cultivated strains of microorganisms isolated only on rich media? Why wasn't a selective medium used to understand the presence of keratinolytic strains in the cultivated microbiota? The same applies to chromium resistance. This would make it possible to more confidently associate the isolated strains with destruction of chromium-tanned leather. Since many isolated strains were identified as bacilli, they could wait out unfavorable conditions in the compost in the form of spores, be metabolically inactive and not contribute to destruction of leather. After being exposed to nutrient media, the bacilli could restore metabolic activity and form colonies. We thank the reviewer for this insightful comment. In this study, the cultivation step was conceived as a first screening stage within a broader research programme, with two main goals: (i) to document the microbial communities associated with chromium-tanned leather under thermophilic composting and (ii) to build a broad collection of cultivable strains from this compost environment. This strain collection is to be tested, together with isolates obtained from other environments, in targeted functional assays such as chromium resistance tests, collagen/keratin degradation assays and standardised leather biodegradation tests. For this reason, we deliberately chose non-selective, nutrient-rich media and incubation conditions favouring fast-growing thermotolerant bacteria, rather than using selective media containing keratin/collagen or chromium at this initial stage. We fully agree that the use of selective substrates or chromium-supplemented media would allow a more direct link between specific isolates and functional traits such as keratinolytic activity or metal tolerance, and that spore-forming bacilli may be over-represented under the conditions we used. As the reviewer correctly points out, these bacilli could survive unfavourable conditions in the compost as spores and resume active growth only on rich laboratory media. For this reason, in the manuscript, we are careful not to claim that all isolated strains are the main in situ agents of leather destruction or chromium transformation; instead, we treat them as candidates of interest for further testing. The aim of the present work is therefore to characterise the microbial context and assemble a representative strain collection, which will be used in subsequent studies specifically designed to assess keratinolytic potential and chromium resistance using dedicated selective media and functional assays, including standardised biodegradation tests.
- In general, little attention has been paid to the description and discussion of cultivated strains. Were 2L and 40L vessels equally good at isolating leather-destroying microorganisms? From which vessels were more strains isolated? Such practical questions are important for those who will conduct similar research. We appreciate the reviewer’s interest in the practical aspects of strain isolation. In this work, however, the cultivation step was not designed to compare the isolation efficiency of the 2 L and 40 L systems, but rather to establish a thermophilic composting system containing chromium-tanned leather that would provide a suitable environment for the recovery of colonising microorganisms. Colonised leather pieces and surrounding compost were sampled from both composting setups and processed together in the isolation workflow. We did not record, for each strain, the specific vessel or scale from which it originated. For this reason, we are not able to provide a robust quantitative comparison of the number of isolates obtained from each vessel size, nor can we conclude that one scale was “better” than the other at yielding leather-associated strains. Nevertheless, we distinguished between isolates obtained from leather fragments and those from the surrounding compost (using different strain prefixes), as we considered the origin (leather vs. compost) more informative than the specific reactor volume for interpreting the role of these strains in the system. More generally, even when composting systems are prepared with the same inputs and operated under nominally similar conditions, local microenvironments, stochastic colonisation events, and slight differences in process history can strongly influence which strains are ultimately recovered on culture media. In our view, this makes it difficult to generalise from our setup to a universal ranking of “best” vessel size for isolation. Instead, the practical information we can offer others wishing to perform similar work is a description of the compost matrix, leather size and exposure conditions, the temperature regime, and the sampling strategy, which together define an experimental framework in which thermophilic, leather-associated strains can be obtained. In the revised version of the manuscript, we have clarified that our goal was to establish and characterise such a framework and to assemble a representative strain collection, rather than to optimise or compare isolation yields between the two composting scales.
Short remark:
In Table 2, Table 3 the numbers of the strains from the GenBank closest to the isolated strains were not indicated. Although this was written in the description to the tables. The mention in the captions of Table 2 and Table 3 that “GenBank accession numbers of the closest strains are indicated” was an oversight, because these accession numbers were not included in the tables. In the revised version of the manuscript, we have corrected this inconsistency by removing that sentence from both table captions.
Round 2
Reviewer 3 Report
Comments and Suggestions for Authors
Dear authors of the manuscript and editors of the journal,
I have reviewed the changes in the article and read the answers and explanations attached to the revised version. In my opinion, the quality of the manuscript has improved as much as the research design allows.